# Ubiquinone Metabolism and Transcription HIF-1 Targets Pathway Are Toxicity Signature Pathways Present in Extracellular Vesicles of Paraquat-Exposed Human Brain Microvascular Endothelial Cells

**DOI:** 10.3390/ijms22105065

**Published:** 2021-05-11

**Authors:** Tatjana Vujić, Domitille Schvartz, Anton Iliuk, Jean-Charles Sanchez

**Affiliations:** 1Department of Medicine, University of Geneva, 1206 Geneva, Switzerland; tatjana.vujic@unige.ch (T.V.); domitille.schvartz@unige.ch (D.S.); 2Swiss Centre for Applied Human Toxicology, 4055 Basel, Switzerland; 3Tymora Analytical Operations, West Lafayette, IN 47906, USA; anton.iliuk@tymora-analytical.com

**Keywords:** extracellular vesicles, endothelial cells, paraquat, CNS, BBB, proteomics, DIA-MS, ubiquinone metabolism, oxidative stress, HIF-1, hypoxia

## Abstract

Over the last decade, the knowledge in extracellular vesicles (EVs) biogenesis and modulation has increasingly grown. As their content reflects the physiological state of their donor cells, these “intercellular messengers” progressively became a potential source of biomarker reflecting the host cell state. However, little is known about EVs released from the human brain microvascular endothelial cells (HBMECs). The current study aimed to isolate and characterize EVs from HBMECs and to analyze their EVs proteome modulation after paraquat (PQ) stimulation, a widely used herbicide known for its neurotoxic effect. Size distribution, concentration and presence of well-known EV markers were assessed. Identification and quantification of PQ-exposed EV proteins was conducted by data-independent acquisition mass spectrometry (DIA-MS). Signature pathways of PQ-treated EVs were analyzed by gene ontology terms and pathway enrichment. Results highlighted that EVs exposed to PQ have modulated pathways, namely the ubiquinone metabolism and the transcription HIF-1 targets. These pathways may be potential molecular signatures of the PQ-induced toxicity carried by EVs that are reflecting their cell of origin by transporting with them irreversible functional changes.

## 1. Introduction

Extracellular vesicles (EVs) are nanosized particles (<100 nm to <1000 nm) bounded by a lipid bilayer and exempt of nucleus that are naturally released by their cell of origin [1,2,3,4]. Due to the known heterogeneity of EVs, their size overlapping and the lack of specific markers to distinguish EV subsets, the minimal information for studies of extracellular vesicles 2018 (MISEV2018) recommended by the International Society for Extracellular Vesicles (ISEV) has not established as strict EV nomenclature. They are distinguished based on their size, namely small EVs (<100 or <200 nm) or/and as large EVs (>200 nm to <1000 nm) [2]. Since EVs carry nucleic acids, lipids, proteins and metabolites, they provide a valuable source of information that can be transferred into recipient cells and act as an “intercellular mediator” [3,4]. However, their composition may be strongly influenced by (patho-) physiological conditions and environmental stimuli of the cell of origin [4,5]. The EV content and delivery in the circulation of the neurovascular unit became an important field of research to understand their role in the physiology and pathologies of the central nervous system (CNS) [3,6,7].

Despite the existing studies published on EVs from a wide range of cell types, including endothelial cells, there is limited information available on EVs from human brain microvascular endothelial cells (HBMECs). Indeed, HBMECs are important, as they are one of the cell types composing the blood-brain barrier (BBB) [8,9]. In the human brain, HBMECs are playing a major role by joining tight junctions that constitute the anatomical foundation of the BBB [8,10]. HBMECs also assure the nutriment and macromolecule exchanges in the BBB [8], representing a crucial physiological interface between the blood and the CNS [11]. Nevertheless, in conditions where BBB is disturbed, for example by inflammation, a stroke or an exposure to a xenobiotic, the protecting role of HBMECs might be compromised [12]. BBB breakdown has already been described and linked to some neurodegenerative diseases such as Alzheimer’s, Huntington’s and Parkinson’s diseases or multiple sclerosis [9,13,14]. Similarly, exposure to environmental toxicants might be a cause for the long-term development of neurodegenerative diseases [15,16].

Paraquat (PQ), a commonly used herbicide, is transported by neutral amino acid transporters in the BBB and has already been demonstrated to have significant neurotoxic effect on the brain [17,18,19,20,21]. This xenobiotic also gained interest in the study of Parkinson’s disease (PD), as its chemical structure is highly similar to the molecular model, 1-methyl-4-phenyl-1,2,3,6-tetrahydropyridine (MPTP), used in Parkinson’s disease research [22,23]. Despite the fact that several studies highlighted the oxidative stress as a major toxic effect of PQ in different brain cell types [17,20,21,24], there is no available information on its effect on HBMEC-released EVs.

In the present study, EVs from primary human brain endothelial cells were characterized by mass spectrometry-based proteomics. The major finding of this study is that EVs exposed for 24 h to paraquat at 100 µM displayed modulation of two biological pathways, which are highly significant: the ubiquinone metabolism and the transcription of HIF-1 targets. These two biological pathways highlight that EVs carried information from their parent cells, as they are two well-described pathways impacted by the PQ.

## 2. Results

### 2.1. Characterization of Extracellular Vesicles from Human Brain Microvascular Endothelial Cells

EVs from primary human brain microvascular endothelial cells were isolated using an EVtrap isolation method based on a chemical affinity capture approach, permitting to isolate small extracellular vesicles [25,26].

Three proteins were selected to confirm EV enrichment by western blots. PDCD6IP (Alix) and TSG101 are cytosolic proteins, whereas the calreticulin (CALR) is localized at the lumen of the endoplasmic reticulum, making it a suitable candidate as a negative enrichment control [27]. As shown in Figure 1A, detection of the PDCD6IP and TSG101 markers indicated a successful isolation of EVs [28]. Calreticulin was only present in whole cell lysate, which indicated low contamination from endoplasmic reticulum or apoptotic bodies in EVs [27].

To further characterize EVs, a proteomic analysis using mass spectrometry was conducted. Overall, a total of 1452 proteins were identified (Appendix A). Typical EV markers, as PDCD6IP and TSG101, were confidently identified by mass spectrometry, confirming previous results.

A deep analysis of the identified proteins in EV samples highlighted that a majority of the proteins (>70%) were annotated as EV proteins in Vesiclepedia, a reference database of extracellular vesicle composition [29]. Proteins from Vesiclepedia were filtered by selecting “*homo sapiens*” for species, “protein” for content types, “endothelial cell” for cell types and “mass spectrometry” for methods on FunRich 3.1.3 [29,30] (Figure 1B). In addition, mass spectrometry identified 85 proteins common to the Top 100 EV markers annotated in the Vesiclepedia database [29].

Moreover, by comparing to the work of Haqqani et al. [11], functional description of the identified proteins highlighted that “transcription and protein synthesis” (25%), “cell structure and motility” (18%), trafficking and membrane fusion proteins (18%) and “heat shock proteins and chaperones” (13%) are protein categories with a higher proportion in the pie chart (Figure 1C), representing key biological processes. Identified proteins are also part of other important functional categories in EVs, such as antigen-presentation proteins (HLA-A and HLA-C), cell adhesion proteins (MFGE8, THBS1 and integrins), cell structure and motility proteins (ACTN, CFL1, TUB, MYH and MSN), heat shock proteins and chaperons proteins (HSP90, HSPA4, HSPA5, HSPA8 and HSPA9), multi vesicular body proteins (PDCD6IP, TSG101, CHMP3 and CHMP4B), signaling proteins (YWAH, ARHGDIA and RHOC), tetraspanin proteins (CD9, CD44 and CD59), trafficking and membrane fusion proteins (annexins and rab proteins) or transcription and protein synthesis (histones and ribosomal proteins) [11] (Appendix A). These proteins are also described as EV markers in other EV proteomic studies, including those from BBB or endothelial cells [11,31,32,33,34,35,36].

To consolidate these results, a bioinformatics analysis was performed on the list of identified EV proteins. Enrichment of cellular component from Gene ontology (GO) ranked terms such as “extracellular exosome” (enrichment *p*-value of 4.05 × 10^−300^), “extracellular vesicle” (enrichment *p*-value of 1.65 × 10^−299^) and “extracellular organelle” (enrichment *p*-value of 5.09 × 10^−299^) (Figure 1D) with highly significant *p*-values, confirming enrichment of EVs from primary brain endothelial cells.

### 2.2. Modulation of Vesicular Protein Content after Paraquat (PQ) Exposure

Before performing EV analyses from primary brain endothelial cells after paraquat exposure, paraquat-induced toxicity was evaluated for 24 h at different paraquat concentrations on HBMECs. Cell proliferation was assessed by MTS Proliferation Assay and cell cytotoxicity by measuring LDH release. No effect was denoted on cytotoxicity or proliferation of HBMECs after PQ exposure at 1, 10 and 100 µM (Appendix A). However, as a short exposure time (24 h) was chosen, it was decided to apply PQ concentration at 100 µM to enhance the biological observation across the whole study.

To determine whether 24 h exposure to PQ at 100 µM alters EV size distribution and concentration (particles/mL), a nanoparticle tracking analysis (NTA) was conducted. The control group presented a mean EV size of 126.70 ± 10.10 nm and the PQ-treated group at 143.90 ± 18.90 nm (Figure 2A), showing a slight significant difference between the two groups (*p*-value = 0.0222). EV concentration was also measured by NTA, resulting in 3.50 × 10^7^ ± 4.00 × 10^6^ particles/mL for the control group and 2.60 × 10^7^ ± 4.20 × 10^6^ particles/mL for the treated one (Figure 2B), indicating a significant difference in number of particles released between the two groups (*p*-value = 0.001). These results suggest that the size and concentration of EVs are impacted by PQ exposure. Additionally, NTA measurement demonstrated that EVs purified from primary endothelial cells (from control and treated group) were in the expected diameter range of small vesicles (<100 or <200 nm), validating the high selectivity of EV preparation [2].

To evaluate proteome modifications in EVs under PQ exposure, a quantitative proteomic analysis was conducted by data-independent acquisition mass spectrometry (DIA-MS). A total of 1452 proteins were quantified of which 107 were found to be differentially expressed (DE) in PQ-treated EVs (│FC│ > 1.2; LFDR ≤ 0.05) (Appendix A). As illustrated in the volcano plot (Appendix A), among the most highly DE proteins of the 107 DE proteins, three (LOXL2, MMP2 and PHB2) are known to have a link with hypoxia response [37,38,39]. This observation permits to denote a previously described effect of the PQ induction of hypoxia-related pathways [40].

Gene Ontology (GO) enrichment analysis was performed, showing that most of the DE proteins were enriched with the highest significance in biological processes, such as GO terms related to oxidative processes (oxidative phosphorylation, oxidation-reduction process) and to mitochondria (ATP metabolic process, mitochondrion organization, electron transport chain) (Appendix A).

Pathway enrichment analysis was also performed using proteins significantly regulated upon PQ exposure. The results highlighted that paraquat modulates pathways of the ubiquinone metabolism (enrichment *p*-value of 2.56 × 10^−6^) and the transcription of HIF-1 targets (enrichment *p*-value of 1.49 × 10^−4^) pathways (Figure 3). Most of the other pathways uncovered were related to the hypoxia or the multifunctional cytokine, the TGF (Figure 3). These results reflect the well-described effect of the PQ as an oxidative stress inducer [18,41,42]. Indeed, enriched pathways found in EVs are corroborating the observation already noticed in HBMECs treated by PQ [43].

## 3. Discussion

In the brain, accumulating studies have demonstrated that EVs are capable to cross the blood-brain barrier (BBB) [44,45], acting inside the CNS and at the periphery, which makes them valuable for biomarker research. However, unlike in other cellular types constituting the BBB (e.g., astrocytes, microglia or pericytes), EVs of human brain microvascular endothelial cells (HBMECs) were poorly characterized. This study aimed at examining the modulation of EV proteome profiles after paraquat (PQ) exposure using proteomics-based strategies.

EVs enriched by EVtrap met required EV physical characteristics according to the MISEV 2018 criteria [2]. Size distribution measured was lower than 200 nm, underlying that EVtrap beads capture a small EV subset, as demonstrated in previous studies [25,26]. EVs were also shown by western blot to contain common well-characterized markers-PDCD6IP and TSG101 [28]. On the other hand, calreticulin was undetectable in EVs, indicating absence of cellular contamination. Moreover, proteins identified by mass spectrometry were shown to belong to different known categories of EV markers, highlighting that EVs captured by EVtrap were consistent with described EV composition. Cellular component enrichment revealed that most of the proteins were related to GO terms linked to “extracellular exosome”, “extracellular vesicle” and “extracellular organelle” with highly significant *p*-values. These results were in accordance with previous proteomic findings [32,46,47], suggesting that EVs were properly isolated and enriched by EVtrap capture.

PQ concentration was chosen based on other studies using this xenobiotic [48,49,50,51] and was verified to be sufficient to withstand a physiological change without inducing a cytotoxic effect in the cell type of this study. As the exposure time to the toxin was relatively short (24 h), the highest concentration was selected. Moreover, a study using other brain cells (astrocytes and neurons) and two toxins (paraquat and rotenone) has reported that both cell type were less sensitive to PQ compared to rotenone. To elicit significant effect by the toxin, they used PQ at 100 µM after 24 h exposure, supporting PQ concentration chosen in our study [48]. To further evaluate EVs as “molecular signature” carriers from HBMECs after 24 h exposure of PQ at 100 µM, NTA analysis was performed. NTA measurement demonstrated a change in size distribution in PQ-treated EVs, which were bigger than the control. On the other hand, concentration (vesicles/mL) post-treatment was statistically lower compared to the control group. A study on microglia-derived extracellular vesicles stimulated with TNF-α reported similar trends by increasing mean size distribution and decreasing EV number [52], which suggests that an external stimulus would have an impact on EV size and concentration. However, other EV studies described the opposite, namely a decrease in size distribution and an increase in EV number after exposure to a stimulus [53]. Due to this controversial observation, it can be assumed that EV size and concentration may vary differently depending on the origin of the stimulus, its concentration, the biological material used or the isolation method chosen [52,53,54,55]. Nonetheless, one may speculate that a change in EV size or concentration would have an influence on their ability to fuse at the plasma membrane of a recipient cell as well as their uptake in the extracellular environment.

The quantitative mass spectrometry analysis of EVs from PQ-exposed HBMECs demonstrated that more than a hundred proteins were significantly modulated by the xenobiotic exposition. A large number of the differentially expressed proteins (DEP) are located in the mitochondria. Similarly, Gene Ontology (GO) terms enrichment highlighted GO terms such as “ATP metabolic processes”, “mitochondrion organization” or “electron transport chain”, which are closely related to the mitochondria. In addition, a quarter of DEP were linked to the oxidative phosphorylation directly associated to the complex I of mitochondria (NDUFA5, NDUFA9, NDUFA13, NDUFS2, NDUFS3, NDUFV1, ATP5H, ATP5O, ATP5J2, STOML2), as well as oxidation-reduction processes. One of the most described effect induced by PQ is the generation of reactive oxygen species leading to altered biological processes related to cellular oxidation [18,21,22]. Moreover, PQ interaction with mitochondria remains the cornerstone of its toxicity mechanism, particularly in the brain [18,24]. These results are in line with our own study exploring the proteome by mass spectrometry of the entire PQ-treated cells [43]. Moreover, it can be presumed that cells stressed by paraquat exposure will result in damage to cellular membranes, permitting part of mitochondria to be released in the extracellular medium. A very recent study of D’Acunzo et al. identified a novel population of EVs of mitochondrial origin altered in Down syndrome [56]. Indeed, these EVs contain a specific subset of mitochondrial proteins [56]. These combined results confirm that altered EV protein content can mirror the molecular status of the parent cells, as demonstrated by other studies [56,57,58,59,60,61,62].

Finally, enrichment pathway analysis of EVs exposed to PQ highlighted “ubiquinone metabolism” and “transcription of HIF-1 targets” as pathways enriched with the highest significance. These results suggest that paraquat modulates mechanisms such as oxidative stress and hypoxia-related pathways in EVs. As described above, PQ is an oxidative stress inducer that inhibits respiratory chain complexes in the mitochondria (especially complex I, which is directly related to the ubiquinone metabolism) [24]. In addition, hypoxia-inducible factor-1 (HIF-1) is a transcription factor sensitive to the oxygen species and regulates cellular response to changes in oxygen tension during normal development or pathologic processes [24]. HIF-1 has already been linked to PQ-induced toxicity in pulmonary fibrosis by increasing its expression [40,63]. Our proteomic profiles of PQ-treated vesicles also demonstrated an action of PQ at the vesicular level by modulating the expression of proteins belonging to the ubiquinone metabolism as well as to the hypoxia downstream mechanisms. These observations are consolidating our assumption that EVs are reflecting PQ effects by carrying pathway signatures originating from parent cells. Likewise, a study on malignant brain tumor glioblastoma multiform concluded that the proteome and mRNA profile of EVs were a mirror of the oxygenation status found in cells and that the hypoxia-dependent intercellular signaling pathway may be a potential targeted driver during tumor development [60]. Nevertheless, the vesicular content, such as mRNAs and lipids, was not explored in this study. This raises interest to further research this topic adding complementary information about oxidative stress and hypoxia mechanisms as “signature pathways” modified by PQ in EVs [64,65]. Moreover, as a follow-up to this study, it would also be interesting to explore the interaction of EVs from primary brain endothelial cells with other components of the neurovascular unit (i.e., astrocytes, pericytes, neurons) in order to evaluate the vesicles–cells communication and to identify the molecules crossing the blood-brain barrier.

In conclusion, a qualitative proteomics study allowed us to confirm the proper isolation and enrichment of EVs from HBMECs. The quantitative portion of this study also provided the first insight into EV proteome profiles from HBMECs after a xenobiotic exposure such as PQ. Results suggested that proteins from EVs shared common modified biological pathways to their parent cell, most notably the ubiquinone metabolism and the transcription HIF-1 targets. Both pathways are mechanisms already reported in PQ-induced toxicity in brain research, suggesting their potential use as “signature pathways” to demonstrate PQ effects on the BBB.

## 4. Materials and Methods

### 4.1. Cell Culture

Primary human brain microvascular endothelial cells (ACBRI 367, Cell Systems) were cultured onto a rat tail collagen type I-coated (15 µg/mL, Merck Millipore, Darmstadt, Germany) flask (T150) and maintained in complete endothelial cell growth medium-2 (EGM-2MV BulletKit, Lonza, Walkersville, Maryland) at 37 °C in a 5% CO_2_ incubator. At 80% confluence, cells were washed three times with phosphate-buffered saline (PBS, Life technologies, Bleiswijk, The Netherlands) solution containing calcium and magnesium. They were incubated at 37 °C with 20 mL of complete endothelial cell growth medium-2 containing 5% of heat-inactivated exosome-depleted fetal bovine serum (Gibco/Life technologies, Bleiswijk, The Netherlands) for 24 h. Three T150 were treated with paraquat (Sigma-Aldrich, St. Louis, MO, USA) (100 µM) for 24 h. Afterwards, the medium was collected and used for the extracellular vesicle (EV) isolation. Cells were detached with Stempro Accutase (Gibco/Life technologies, Bleiswijk, The Netherlands) and washed three times with ice-cold Phosphate Buffered Saline (PBS, Gibco/Life technologies, Bleiswijk, The Netherlands), pelleted and dry-stored at −80 °C.

### 4.2. EV Isolation

In each T150 flask, about 10 million of human brain endothelial cells were cultured. 20 mL of cell media were recovered. Cells and apoptotic bodies were removed using centrifugation (2000 g, 20 min). EVs were isolated with 1 mL of cell supernatant by using EVtrap magnetic beads provided by Tymora Analytical as a suspension in water.

The media loading buffer were added at 1:10 *v*/*v* ratio of the cell supernatant. 24 μL of EVtrap magnetic beads were added. The samples were incubated by end-over-end rotation for 1 h, according to manufacturer’s instructions. Following supernatant removal using a magnetic separator rack, the beads were washed with PBS and the EVs eluted by two 10-min incubations with 100 mM of fresh elution solution. Both eluted EVs were pooled and either resuspended in 80 μL of 0.2 μm-filtered water for nanoparticle tracking analysis (NTA) or dry-stored at −80 °C.

### 4.3. Nanoparticle Tracking Analysis (NTA)

NTA was carried out using a Particle Metrix ZetaView® instrument (Particle Metrix GmbH, Inning, Germany). EVs were diluted at 1/500 with 0.2 μm-filtered PBS prior to analysis. To evaluate the total particle count and the overall size, the samples were measured in scatter mode using the 488 nm and standard instrument settings (sensitivity: 80, shutter: 100, min. brightness: 30; min. area: 10; max area: 1000). The samples were measured with ZetaView^®^ software version 8.05.12 SP1 (Particle Metrix GmbH, Inning, Germany).

### 4.4. Protein Extraction and Quantificatio

Cell and EV pellets were resuspended in 80 μL of 0.1% Rapigest (Waters, Milford, MA, USA) 100 mM TEAB (Sigma-Aldrich, St. Louis, MO, USA), incubated 10 min at 80 °C and then sonicated (five cycles of 20 s with breaks on ice). Samples were then spun down (14,000 g, 10 min, 4 °C) and the supernatant was recovered. Protein content was measured using the Bradford assay (Bio-Rad, Hercules, CA, USA).

### 4.5. Western Blot Analysis

The equivalent of 10 μg of proteins for cell samples and for EV samples were separated using electrophoresis on a 10% T/2.6% C polyacrylamide gel and were subsequently transferred onto a PVDF membrane. Membranes were stained with Amido black to highlight the proteins and washed with water to remove the excess. Immunoblot assays were performed using an anti-mouse antibody against PDCD6IP at a dilution of 1:500 (Biolegend, San Diego, CA, USA), anti-rabbit TSG101 at a dilution of 1:500 (Abcam, Cambridge, UK) and anti-rabbit calreticulin (negative marker) at a dilution of 1:100 (Abcam, Cambridge, UK).

### 4.6. Sample Preparation for Mass Spectrometry-Based Proteomics

For each sample, 5 µg proteins was reduced using TCEP (final concentration 5 mM, 30 min, 37 °C) (Sigma-Aldrich, St. Louis, MO, USA), alkylated using iodoacetamide (final concentration 15 mM, 60 min, RT, in dark condition) (Sigma-Aldrich, St. Louis, MO, USA) and digested by an overnight tryptic digestion (*w*/*w* ratio 1:50) (Promega, Madison, WI, USA). The RapiGest surfactant was cleaved by incubating samples with 0.5% trifluoacetic acid (Sigma-Aldrich, St. Louis, MO, USA) (45 min, 37 °C). Samples were then desalted on a C18 reverse phase column (Harvard Apparatus, Holliston, MA, USA), peptides were dried under vacuum and subsequently resuspended in 5% ACN 0.1% FA (peptides final concentration of 0.5 µg/µL and spiked with iRT peptide (Biognosys, Schlieren, Switzerland) (1:20)).

### 4.7. Data Independent Acquisition Mass Spectrometry (DIA-MS) and Data Analysis

For each sample (cells and EVs), the equivalent of 2 µg of peptides were analyzed using Liquid Chromatography–Electrospray ionization-MS/MS (LC-ESI-MS/MS) on an Orbitrap Fusion Lumos Tribrid mass spectrometer (Thermo Fischer Scientific) equipped with an EASY nLC1200 liquid chromatography system (Thermo Fisher Scientific). Peptides were trapped on a 2 cm × 75 μm i.d. PepMap C18 precolumn packed with 3 μm particles and 100 Å pore size. Separation was performed using a 50 cm × 75 μm i.d. PepMap C18 column packed with 2 μm and 100 Å particles and heated at 50 °C. Peptides were separated using a 160-min segmented gradient of 0.1% formic acid (solvent A) and 80% acteonitril 0.1% formic acid (solvent B) at a flow rate of 250 nl/min, following the published protocol [66]. Data-Independent Acquisition (DIA) was performed with MS1 full scan at a resolution of 60,000 (FWHM) followed by 30 DIA MS2 scan with variable windows. MS1 was performed in the Orbitrap with an AGC target of 1 × 10^6^, a maximum injection time of 50 ms and a scan range from 400 to 1250 m/z. DIA MS2 was performed in the Orbitrap using higher-energy collisional dissociation (HCD) at 30%. Isolation windows (30) were variables with an AGC target of 2 × 10^6^ and a maximum injection time of 54 ms. The raw DIA MS data were matched against the spectral library following the published protocol [66].

For the analysis of cells and EVs, protein abundances were exported from Spectronaut™, and selected proteins were tested for significance using Student’s two-tailed t-test. Proteins and peptide intensities were exported and analyzed using mapDIA. No further normalization was applied. The following parameters were used: min peptides = 2, max peptides = 10, min correl = −1. min DE = 0.01, max DE = 0.99 and experimental design = replicate design. Proteins were considered to have significantly changed in abundance with a LFDR < 0.05 and an absolute fold change (|FC|) > 1.2. Data are available via ProteomeXchange with identifier PXD024691.

### 4.8. Enrichment Pathway Analysis

The list of differentially expressed proteins was then analyzed with MetaCore™ version 21.1 (Clarivate Analytics, Philadelphia, USA) to highlight significantly represented biological pathways. Top 10 biological pathways were selected.

### 4.9. MST Proliferation and LDH Cytotoxicity Assay

HBMEC were seeded in a 96-well plate (10,000 cells per well) and treated for 24 h with PQ at different concentrations (0.1, 1, 10, 100, 500 and 1000 µM). Cell proliferation was determined using the MTS assay (CellTiter 96^®^ AQueous One Solution Cell Proliferation Assay, Promega, Madison, WI, USA), whereas cytotoxicity was assessed by measuring lactate dehydrogenase (LDH) released using a Pierce™ LDH cytotoxicity kit (Thermo Scientific, Rockford, IL, USA). Both the MTS and LDH assays were performed according to the manufacturer’s recommendations.

### 4.10. Statistical Analysis

Data are reported as mean ± standard deviation (SD). *p* < 0.05 was considered as statistically significant. Significance is denoted as * *p* < 0.05, ** *p* < 0.01, *** *p* < 0.001, **** *p* < 0.0001. The data were analyzed using multiple t-test comparisons or one-way analysis of variance (ANOVA).

## Figures and Tables

**Figure 1 ijms-22-05065-f001:**
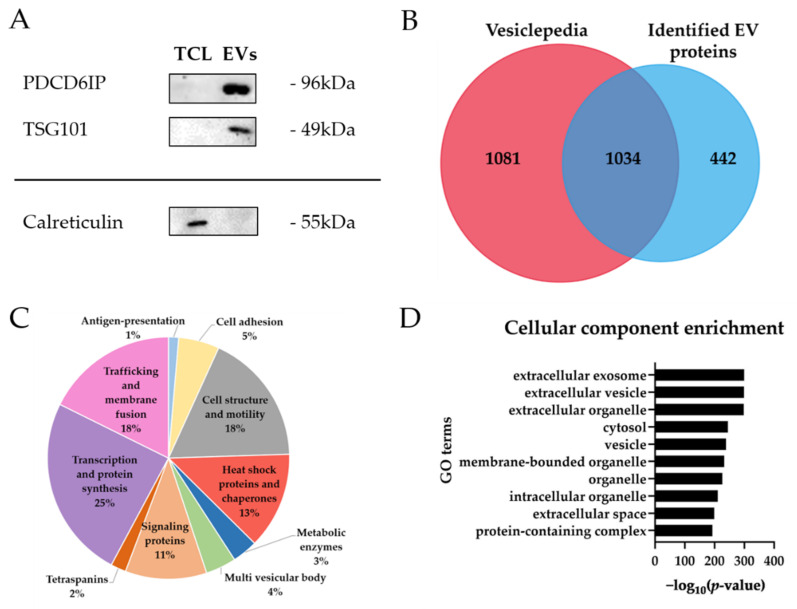
Characterization of extracellular vesicles from primary human brain microvascular endothelial cells (HBMECs). (**A**) Western blot analysis of common EV markers (PDCG6IP, TSG101) and cell organelle marker (calreticulin) in total cell lysate (TCL) and EVs. (**B**) Venn diagram of EV proteins extracted from Vesiclepedia (left) and identified proteins in HBMEC-released EVs (right). (**C**) Pie chart repartition of identified proteins in HBMEC-released EVs into EV functional categories. (**D**) Top ten cellular component enrichment of identified proteins in HBMEC-released EVs. *X*-axis corresponds to −log_10_ (*p*-value), *Y*-axis corresponds to the GO terms. The *p*-value cut-off is set at 0.05.

**Figure 2 ijms-22-05065-f002:**
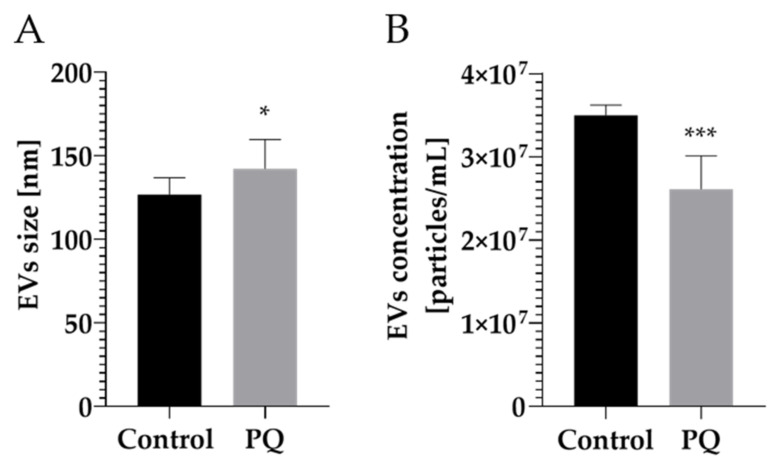
Nanoparticle tracking analysis (NTA) of isolated EVs from primary human brain microvascular endothelial cells in control and treated conditions. (**A**) Size distribution of HBMEC-released EVs in control and treated conditions. (**B**) Concentration of HBMEC-released EVs in control and treated conditions. All data were expressed as mean ± SD. * corresponds to *p*-value ≤ 0.05 and *** corresponds to *p*-value ≤ 0.0005.

**Figure 3 ijms-22-05065-f003:**
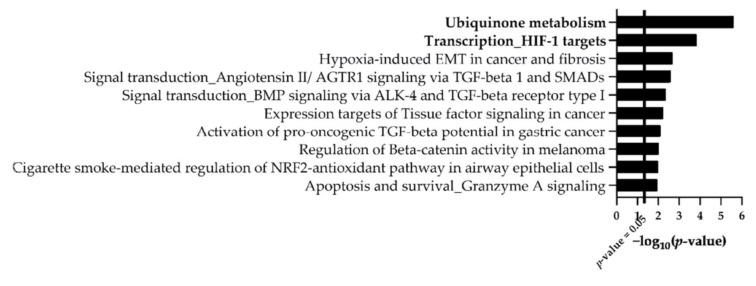
Top ten enriched biological pathways provided by MetaCore™ software for the lists of changing proteins (|FC| > 1.2, *p*-value ≤ 0.05, *n* = 3) after PQ treatment on EVs from HMBECs. *X*-axis corresponds to −log_10_ (*p*-value), *Y*-axis corresponds to the biological pathways and the vertical line represents the *p*-value cut-off of 0.05.

## Data Availability

Not applicable.

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
