# Peer review of "Ubiquinone Metabolism and Transcription HIF-1 Targets Pathway Are Toxicity Signature Pathways Present in Extracellular Vesicles of Paraquat-Exposed Human Brain Microvascular Endothelial Cells"

_ijms, 2021, doi:10.3390/ijms22105065_

Round 1

Reviewer 1 Report

Vujic et al submit an original research paper entitled "Ubiquinone metabolism and Transcription HIF-1 targets pathway are toxicity signature pathways present in Extracellular Vesicles of Paraquat-exposed Human Brain Microvascular En-4 dothelial Cells". In this interesting paper, they isolated and characterized EVs from human brain microvascular endothelial cells (HBMECs), after paraquat (PQ) stimulation. They show that EVs exposed to PQ have modulated pathways, ubiquinone metabolism and HIF-1 .

Cells : what do the authors mean by "EVs from primary human brain microvascular endothelial cells were isolated using an EVtrap isolation method based on a chemical affinity capture approach [25, 26]. Because of the current discrepancy around the definition of EV subsets, this study will not distinguish EV subtypes but will describe EVs as a single entity [2, 25]."

From their mat and meth they declare that cells were provided by a supplier"Primary human brain microvascular endothelial cells (ACBRI 367, Cell Systems)"

What do the authors know about heterogeneity of the material: different donnors, pathologies of the donnors etc?

The authors decided to apply PQ concentration at 100μM. Is this consistent with in vivo concentrations in human serum ?

a recapitulative figure would be welcome.

Minor

line 34. Rewrite "They are named based on their size as"

Author Response

Response to Reviewer 1 Comments

Point 1: what do the authors mean by "EVs from primary human brain microvascular endothelial cells were isolated using an EVtrap isolation method based on a chemical affinity capture approach [25, 26]. Because of the current discrepancy around the definition of EV subsets, this study will not distinguish EV subtypes but will describe EVs as a single entity [2, 25]."

Response 1: We thank the reviewer for pointing out this comment. The paragraph was modified in the manuscript at the lines: 84-86.

Point 2: From their mat and meth they declare that cells were provided by a supplier"Primary human brain microvascular endothelial cells (ACBRI 367, Cell Systems)"

What do the authors know about heterogeneity of the material: different donnors, pathologies of the donnors etc?

Response 2: We thank the reviewer for raising this point. The Primary human brain microvascular endothelial cells (ACBRI 367) provided by Cell Systems are from multidonor pool and isolated from normal/heathly donor tissues according to the information that we requested from Cell System support.

Point 3: The authors decided to apply PQ concentration at 100μM. Is this consistent with in vivo concentrations in human serum ?

Response 3: We thank the reviewer for raising this point. Results of a recent paper from Yuan, G, Li, R, Zhao, Q, et al. highlighted that blood concentration of patients after paraquat acute poisoning ranged from 0.10 to 20.62 μg/mL, which correspond to 0.38 to 80 µM. However, our study was performed in an in vitro monoculture cell. In addition, interaction with other components of the neurovascular unit (i.e. astrocytes, pericytes, neurons) was not considered but could also modified paraquat concentration as well as its transportation. Nonetheless, this concentration has been used in several studies in in vitro and in vivo models (line 208 in the manuscript) and has no impact on cytotoxicity or proliferation as shown in our experiments.

Point 4: A recapitulative figure would be welcome.

Response 4: We thank the reviewer for suggesting a recapitulative figure, we added the figure in the manuscript just below the abstract as a graphical abstract. Please, do no hesitate to ask for more details if it is not enough.

Point 5: Minor line 34. Rewrite "They are named based on their size as"

Response 5: We modified the sentence in the manuscript (line 35-37).

Reviewer 2 Report

The manuscript presented by Tatjana Vujic and et al. titled “Ubiquinone metabolism and Transcription HIF-1 targets pathway are toxicity signature pathways present in Extracellular Vesicles of Paraquat exposed Human Brain Microvascular Endothelial Cells” is well written, clear, and easy to read. It adds information to the subject area of human brain research.

The Extracellular vesicles (EVs) is a frontier topic in several fields.

Please adds a colored picture in figure 1

Author Response

Response to Reviewer 2 Comments

Point 1: Please adds a colored picture in figure 1

Response 1: We thank the reviewer for this comment. Figure 1 was modified as suggested by the reviewer.

Round 2

Reviewer 1 Report

changes are ok